# Descriptive Study of Friendship Networks in Adolescents and Their Relationship with Self-Efficacy and Alcohol Consumption Using Social Network Analysis

**DOI:** 10.3390/ijerph191811527

**Published:** 2022-09-13

**Authors:** Enedina Quiroga-Sánchez, Alberto González García, Natalia Arias-Ramos, Cristina Liébana-Presa, Pilar Marques-Sánchez, Lisa Gomes

**Affiliations:** 1SALBIS Research Group, Faculty of Health Sciences, Ponferrada Campus, University of León, 24401 Ponferrada, Spain; 2Department of Nursing and Physiotherapy, Faculty of Health Sciences, Vegazana Campus, 24071 Ponferrada, Spain; 3School of Nursing, Minho University, 4704-553 Braga, Portugal

**Keywords:** alcohol drinking, adolescent, self-efficacy, social network analysis, friendship

## Abstract

Alcohol is a widely used legal drug. Self-efficacy takes on great importance in an adolescent’s development. Levels of self-efficacy can contribute to the decision-making process of the adolescent. In this phase, a group of friends becomes increasingly important. Alcohol is present in different social aspects. Adolescents use alcohol as a social facilitator or as part of the development of their autonomy. The purpose of this study is to describe and analyze the structure of the friendship network, self-efficacy, and alcohol consumption in adolescents. Methods: We used an online platform with validated self-reported questionnaires to collect the data. The sample consisted of 195 adolescents aged between 16 and 18, from different educational centers in Bierzo (Spain). Results: It was found that 43.6% of the adolescents in our research were high-risk consumers. Significant differences were obtained in terms of self-efficacy and different network metrics. These networks were represented by considering the levels of alcohol consumption. In the friendship network, we observed how the central adolescents presented higher levels of self-efficacy and lower alcohol consumption. Conclusions: Self-efficacy is established as a predictor of coping and not consuming alcohol and adolescence as a crucial moment to implement prevention strategies. Social Network Analysis is a useful tool that provides an image of the context in which adolescents find themselves.

## 1. Introduction

According to the World Health Organization (WHO), alcohol is the most widely used legal drug among young people. From 2002 to 2019, the prevalence of alcohol use in the previous 30 days decreased by 41.1% for youth between 16 and 17 years old. However, alcohol consumption remains an important public health problem [1].

In Spain, the most recent data indicate that alcohol is the most consumed psychoactive substance among secondary school students aged 14 to 18, with an average age of onset of 14 years old for both sexes, this being the age of the first drunkenness, and weekly alcohol consumption around the age of 15. In 2022, more than 340,000 students started drinking alcohol, which represents 48.9% of those who had not started before [2].

Alcohol consumption is integrated into social norms. Young people view this consumption as a normal activity, typical of their age and the group with which they mingle [2]. Alcohol consumption is related to multiple risk factors. The abusive and early intake of alcohol can interfere with the development of adolescents, causing negative consequences in their physical and psychological health. Therefore, the use and abuse of alcohol is one of the main risk factors for health, where adolescent consumers are more likely to show social and personal problems and less psychological adjustment [3,4].

Adolescence is a time of change, a stage of biological, psychological, and social development, a vital stage, a transition between childhood and adulthood, key in the consolidation, gain or loss of habits and lifestyles previously acquired, and that affect the future state of health of people [5,6]. During adolescence, one of the most important personal factors for balanced development is self-efficacy [7]. Self-efficacy refers to the interaction of the individual with the social environment, allowing self-realization and the development of the necessary skills to face the outside world, as well as the acquisition of personal efficacy [8].

Self-efficacy problems in adolescents have been seen as mediators in the consumption of alcohol and other drugs [7]. An adolescent with a low level of self-efficacy has a higher need for approval, so is strongly motivated to follow those behaviors that their peers show in order to strengthen their integration among them. He or she has a low level of confidence in handling risk behaviors, so the pressure of social interaction makes him or her a perfect target for alcohol consumption [8]. Most adolescents agree that they view those who drink alcohol as “fun-loving” and “cool”, as well as happy and well-adjusted [7]. Therefore, adolescent self-efficacy plays a key role in the prevention of the consumption of alcohol and other substances [9,10].

As mentioned above, alcohol is present in many and varied social aspects [3], and adolescents use alcohol as a social facilitator or as part of the development of their autonomy [4]. There are various cognitive models based on self-control that are significantly associated with drug abuse [11]. The social cognitive theory of learning provides an explanation for alcohol consumption and urges the concept of self-efficacy as the most important explanatory component related to the acquisition, maintenance, and behavioral changes in substance abuse [12]. Following this axis, developmental theories consider adolescence as a social construction. In this period, interactions between peers increase, new traits, new roles and social experiences are adopted that mark their personality, and the motivation of adolescents to develop a stable sense of their own identity based on the confidence perceived by their peers arises [13].

In this context, the concept of Social Network Analysis (SNA) emerged, which is showing a significant impact on health matters. An adolescent’s group of friends is an essential element to understand the attitudes and behaviors of the adolescent. The group of friends progressively replac3 the family as a reference, and the most important relationships of the adolescent move towards these friends with similar ages or interests. [14,15]. In recent decades, the SNA has been used in areas such as public health, social support, or influences on health behaviors. Various investigations have used the concepts of centrality within the social structure to reflect how adolescents who consume alcohol tend to have a greater number of social connections than those who do not, being more central and nominated within the network [16,17].

The SNA is a theoretical and methodological paradigm, a formal method by which to measure social relationships and, therefore, the social behavior of individuals in a given environment. The literature has reflected how having friends or being connected to certain friendship networks that exhibit risk behaviors, such as smoking, consuming alcohol or cannabis, etc., implies a higher risk of engaging in these behaviors. Based on this, the SNA is useful to understand these contexts of interaction and determine certain risk behaviors [17]. Adolescent-to-adolescent contacts are important not only for the establishment of the adolescent’s health, but also for the acquisition and maintenance of risk behaviors [18,19].

For all the above, this research states the following question: what is the relationship between the structural characteristics of self-efficacy networks and alcohol consumption in adolescents? To answer this question, the SNA is presented as a useful tool that will allow us to understand the behavior pattern of the network. This way, specific and effective interventions can be established and planned based on the identified risk behaviors. Therefore, the objective of this study is to describe the structure of the friendship network, self-efficacy, and alcohol consumption of adolescents between 14 and 18 years of age and to analyze the relationships between the structural variables of the network.

## 2. Materials and Methods

A cross-sectional descriptive study was carried out. A non-probabilistic sample was selected for convenience. The selection of the centers was carried out considering their receptivity to the project. It was carried out with the consent of the project of all participating schools The adolescents interested in collaborating in our research participated voluntarily after signing an informed consent. The sample consisted of 195 adolescents between the ages of 16 and 19 who attended public schools in the Bierzo area (León, Spain) during the academic year 2017/2018.

### 2.1. Variables and Measurement Instruments

Self-efficacy: Self-efficacy was measured using the Baessler and Schwarzer General Self-Efficacy Scale [20] validated in Spanish [21]. This questionnaire is composed of 10 items with a four-point Likert-type scale (incorrect, hardly true, rather true, and true). The higher the score, the higher the self-efficacy. From this, an average variable of the scores of the different items was generated and dichotomized through the classification by percentiles, thus defining low, medium, and high self-efficacy.

Alcohol consumption: The Alcohol Use Disorders Identification Test (AUDIT) was used to measure alcohol consumption [22]. This questionnaire is made up of ten questions that explore three domains: risky consumption, dependence symptoms, and harmful consumption, using the cut-off point at 8 for males and 6 for females. AUDIT is a sensitive test (51–97%) to detect harmful alcohol use, abuse, or dependence. Currently, it is the most recommended risk screening technique, based on the simplicity of its application and its focus on the recent past [23].

Networks: To establish the friendship network in the classroom, a list of participating students was used for each network established in each class. A network code question was formulated [24,25,26] in which students were asked to nominate those classmates with whom they share their free time, using a 4-point Likert-type scale, where 0 means “I never share my free time” and 4 means “we are always together”. An initial matrix was built from which we obtained 3 adjacency matrices based on the dichotomization by contact intensity. Finally, we obtained 3 matrices of minimum, intermediate and maximum contact intensity.

### 2.2. Ethical Considerations

All personal information that could identify the students was immediately encrypted with a fictitious name by the online questionnaire platform used to guarantee the anonymity and confidentiality of the research participants. The study was approved by the University Ethics Committee (ETICA-ULE-003-2015). Permission was requested from the Ministry of Education of the Junta de Castilla y León. Prior authorization was obtained from their parents or legal guardians to participate in the study, as well as the informed consent of the participants. The data obtained from the investigation was treated in accordance with both the Organic Law 3/2018, of December 5, on the Protection of Personal Data and Guarantee of Digital Rights, and the General Data Protection Regulation of the European Union EU 2016/679 (GDPR).

### 2.3. Data Analysis

To analyze the data, the STATA 14.0 program (StataCorp LLC., 4905 Lakeway Drive, College Station, TX, USA) was used. Qualitative variables were shown as frequencies and percentages. Quantitative variables were expressed as mean and standard deviation. After verifying that the quantitative variables did not follow a normal distribution, using the Kolmogorov-Smirnov test with Lilliefors correction, the Chi-Square formula was applied to the investigation.

The relationships were evaluated using the UCINET 6649 program and NetDraw [27]. Measures of centrality were calculated for the participants. We performed a descriptive analysis of the matrix data, where the results were the values for the degree of connection surrounding each individual (degree), the received connections (input degree) and the given connections (output degree), the degree of proximity (closeness of entry/exit), the degree of intermediation (intermediation), and the level of prestige or influence (eigenvector).

All parameters were analyzed based on a 95% confidence interval and *p* < 0.05 indicated significant values.

## 3. Results

The sample consisted of a total of 195 adolescents, of whom 53.90% (n = 105) were females and 46.10% (n = 90) males, with a mean age of 17 years old (SD = 0.82). A total of 72.30% (195/270) of the students from the participating centers participated in the study. The rest of the students did not give consent for participation in this investigation.

Regarding the results of alcohol consumption measured by applying the AUDIT questionnaire, it was shown that 43.60% of the students (n = 85) were high-risk consumers, and 86.70% of the students stated that they had consumed alcohol at some time in their lives, compared to only 13.30% who had never tried alcoholic beverages. The mean age of starting alcohol consumption was 13.4 years old (SD = 0.67), with the mean age of starting consumption slightly lower in males (13.2 years old; SD = 1.70) than in females (13.6 years old; SD = 1.40).

The descriptive results obtained for the self-efficacy variables are shown in Table 1.

Table 2, Table 3 and Table 4 show the values of the structure of the friendship network in relation to the reflected values of self-efficacy. We observed that the medium-high level of self-efficacy was positively related to the measures of centrality collected within the intermediate and maximum contact intensities.

Figure 1 and Figure 2 represent the friendship networks in the classroom, where those adolescents with high-risk consumption are represented in red and low risk consumers in green. Adolescents with high and medium levels of self-efficacy are identified as triangular and square nodes, respectively, while those with low levels are identified as a circle. The size of the nodes varies according to their total tension.

According to this, those students who present a medium-high level of self-efficacy in the intermediate contact network were related to greater closeness and high levels of indegree (nominations received). We found similar values in the maximum contact network, in which those adolescents with a medium-high level of self-efficacy were associated with having a greater degree of relationships or contacts in the network (degree), with being intermediaries (betweenness), and with positioning themselves as closest figures (closeness), and of greater prestige (eigenvector) with the rest of the network. Thus, we found that individuals who were not consumers of alcohol had higher levels of self-efficacy, were closer and could be considered friends

## 4. Discussion

Although the numbers show the consumption of alcohol has been in decline in recent years, our results reflect how risky consumption has increased. The importance of this issue is mainly due to the change in the pattern of the models of consumption by adolescents. The social, economic, and cultural transformations that continually emerge in today’s society have caused a distortion in the adolescent models. The normalization of alcohol consumption and its connection with the social sphere give adolescents a perception of the absence of risk in relation to consumption [28].

The objective of this article was to describe the friendship network of adolescents and relate it to their self-efficacy and alcohol consumption. We made a representation of the friendship network based on those variables of our research in which significant results have been obtained.

Using the SNA allowed us to locate those individuals with more influential positions within the network. Although there are multiple factors that influence alcohol consumption, the literature indicates that adolescents with higher self-efficacy have greater positivity for social relationships (friendship networks), translating into close and trustworthy people for their peer group [29]. This positive approach is perceived as more effective and resilient to risky behaviors, such as alcohol intake, achieving constancy in the fulfillment of its purposes, strengthening the self-control and confidence to alter and intervene in those situations that generate discomfort or dissatisfaction. The adolescent with a high level of self-efficacy develops close friendships in which a climate of relaxation is created. The ease for social interaction rejects the consumption of alcohol and other drugs among the closest contacts [30,31].

During adolescence, young people replace family by friends of similar ages and interests as a reference element [13]. In this line, the SNA helps us to know the structure of the friendship network in which the adolescent is immersed. In this case, through the SNA, we have identified the position that those adolescents with a high level of self-efficacy maintain within their closest social environment. With our research, we make important contributions to the investigation of the SNA linking self-efficacy and risky alcohol consumption in adolescents, so that adolescents with high self-efficacy maintain central positions with their environment that facilitate the development of positive attitudes and healthy habits.

The literature highlights the importance of prevention and treatment before adolescents develop harmful habits [32]. Adolescents are characterized by being highly influenced by their social environment in a bidirectional way. The individual influences their contacts and these, in turn, influence the individual. Since the habits acquired by adolescents can have an impact on their adult life, it is necessary to quantify these influences and be able to address this problem effectively. Building adolescents with high self-efficacy facilitates social independence and creates more assertive young people with an important active role in the social group in which they are immersed. Self-efficacy is therefore established as an important mediator of facing and withstanding alcohol consumption, contributing to the development of the adolescent’s capacities to resist the pressure of the social environment [33].

This study highlights the importance of SNA as a method. Knowing how adolescents behave and their characteristics in their peer network could provide solutions to establish collective strategies and enable ways to fight this significant public health problem [30]. However, this research has some limitations that must be considered. Self-efficacy and alcohol consumption were measured by self-reports. Therefore, adolescents could have responded with a socially desirable approach. Furthermore, this study was cross-sectional in nature, so causal inferences cannot be made. The small sample size did not allow us to achieve sufficient statistical power. Therefore, the generalization of the results must be done with caution. Likewise, since we cannot guarantee the representativeness of our sample, we cannot ensure that the results obtained have adequate external validity. However, our results agree with other studies [30,31].

In future studies, it would be useful to carry out longitudinal designs that can indicate how these variables behave over time and to carry out causal explanations, as well as designs where health education interventions are proposed to these population groups.

It would be interesting to address self-efficacy in the different dimensions of which it is made up, and not only globally. The findings of this work emphasize the importance of fully investigating self-efficacy to provide richer data on alcohol consumption

## 5. Conclusions

The ARS is a useful tool to understand the social patterns of adolescents. Network key knowledge could facilitate the planning of multifactorial strategies of the environment and the development of preventive strategies that reduce the negative impact of alcohol on young consumers, not only regarding their individual health, but also in the social and family environments, helping to reduce the impact of these problems on the social and health system. The use of the ARS allows us to explain or predict the variables that influence the risky consumption of alcohol in adolescents, being decisive for future actions that guarantee better professionals and quality of life conditions.

## Figures and Tables

**Figure 1 ijerph-19-11527-f001:**
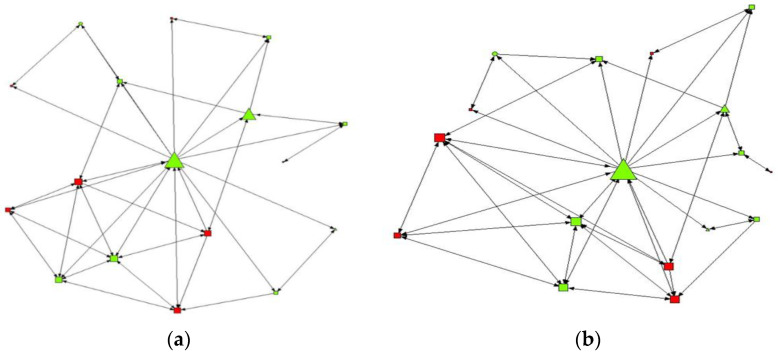
Structure of the friendship network and the study variables (i) Alcohol consumption and (ii) Self-efficacy. (**a**) Intermediate contact intensity by degree of betweenness, (**b**) Intermediate contact intensity by degree of closeness.

**Figure 2 ijerph-19-11527-f002:**
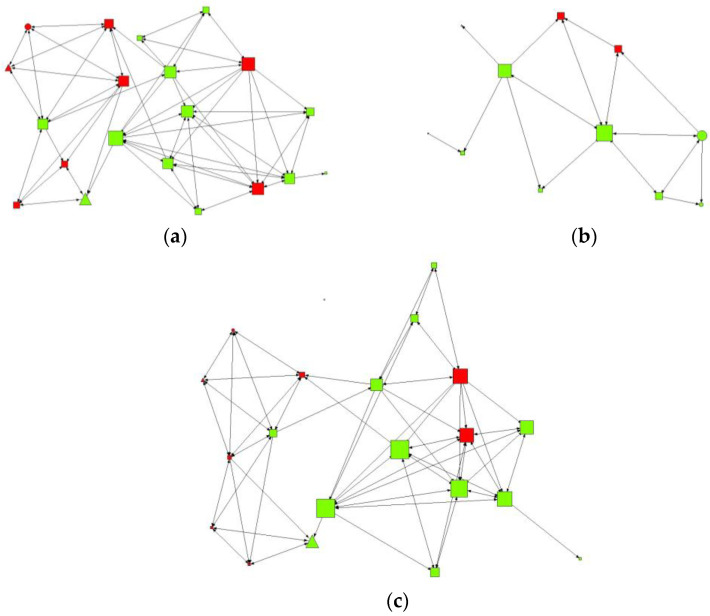
Structure of the friendship network and the study variables (i) Alcohol consumption and (ii) Self-efficacy. (**a**) Maximum contact intensity by degree of intermediation, (**b**) Maximum contact intensity by degree of closeness (**c**) Maximum contact intensity by degree of prestige/influence.

**Table 1 ijerph-19-11527-t001:** Descriptive statistics of self-efficacy.

Questionnaire		n	%
Baessler and Schwarzer Self-Efficacy Scale	Low	24	12.3
	Medium	141	72.3
	High	30	97.4

**Table 2 ijerph-19-11527-t002:** Centrality indicators in relation to self-efficacy. (In Minimum contact intensity).

			Self-Efficacy Level							
			Low		Medium		High		Chi Square	*p*
			n	%	n	%	n	%		
Minimum contact intensity	*outdegree*	1tertile	14	21.2	41	62.1	11	16.7	12.9	0.11
		2tertile	8	12.5	44	68.7	12	18.7		
		3tertile	2	3.1	56	86.1	7	19.8		
	*indegree*	1tertile	14	21.5	41	63.1	10	14.4	13.91	0.08
		2tertile	8	12.5	43	67.2	13	20.3		
		3tertile	12	3.1	57	20.3	7	10.6		
	*degree*	1tertile	14	21.5	41	63.1	10	14.4	13.91	0.08
		2tertile	8	12.5	43	67.2	13	20.3		
		3tertile	12	3.1	57	20.3	7	10.6		
	*incloseness*	1tertile	14	21.5	41	62.1	11	16.7	12.9	0.11
		2tertile	8	12.5	44	68.7	12	18.8		
		3tertile	2	3.1	56	86.1	7	10.8		
	*outcloseness*	1tertile	14	21.5	41	63.1	10	14.4	13.91	0.08
		2tertile	8	12.5	43	67.2	13	20.3		
		3tertile	12	3.1	57	20.3	7	10.6		
	*betweenness*	1tertile	14	21.5	41	63.1	10	14.4	13.91	0.08
		2tertile	8	12.5	43	67.2	13	20.3		
		3tertile	12	3.1	57	20.3	7	10.6		
	*eigenvector*	1tertile	14	21.5	41	63.1	10	15.4	13.9	0.08
		2tertile	8	12.5	43	67	13	20.3		
		3tertile	2	3.1	57	86.4	7	10.6		

Note: degree (degree of relationships that surround each individual), in/outdegree (received and given relationships), in/out closeness (degree of closeness), betweenness (degree of intermediation), eigenvector (degree of prestige or influence).

**Table 3 ijerph-19-11527-t003:** Centrality indicators in relation to self-efficacy. (In Intermediate contact intensity).

			Self-Efficacy Level							
			Low		Medium		High		Chi Square	*p*
			n	%	n	%	n	%		
Intermediate contact intensity	*outdegree*	1tertile	13	20	40	61.54	12	18.46	7.9	0.09
		2tertile	6	9.23	48	73.85	11	16.92		
		3tertile	5	7.69	53	81.54	7	10.77		
	*indegree*	1tertile	14	20.59	42	61.76	12	17.65	11.22	0.02
		2tertile	7	10	50	71.43	13	18.57		
		3tertile	3	5.26	49	85.96	5	8.77		
	*degree*	1tertile	14	20.29	43	62.32	12	17.39	8.4	0.07
		2tertile	5	8.06	46	74.19	11	17.74		
		3tertile	5	7.81	52	81.25	7	10.94		
	*incloseness*	1tertile	14	21.21	40	60.61	12	18.18	10.1	0.04
		2tertile	5	7.94	47	74.60	11	17.46		
		3tertile	5	7.58	54	81.82	7	10.61		
	*outcloseness*	1tertile	14	21.21	40	60.61	12	18.18	10.1	0.04
		2tertile	5	7.94	47	74.60	11	17.46		
		3tertile	5	7.58	54	81.82	7	10.61		
	*betweenness*	1tertile	13	20	40	61.54	12	18.46	8.2	0.08
		2tertile	6	9.38	47	73.44	11	17.99		
		3tertile	5	7.58	54	81.82	7	10.61		
*eigenvector*		1tertile	13	20	40	6.54	12	18.46	8.6	0.07
		2tertile	6	9.68	45	72.58	11	17.74		
		3tertile	5	7.35	56	82.35	7	10.29		

Note: degree (degree of relationships that surround each individual), in/outdegree (received and given relationships), in/out closeness (degree of closeness), betweenness (degree of intermediation), eigenvector (degree of prestige or influence).

**Table 4 ijerph-19-11527-t004:** Centrality indicators in relation to self-efficacy. (In Maximum contact intensity).

			Self-Efficacy Level							
			Low		Medium		High		Chi Square	*p*
			n	%	n	%	n	%		
Maximum contact intensity	*outdegree*	1tertile	14	20.29	39	56.52	16	23.19	19.4	0.001
		2tertile	2	3.33	47	78.33	11	18.33		
		3tertile	8	12.12	55	83.33	3	4.55		
	*indegree*	1tertile	14	19.44	42	58.33	16	22.22	16.3	0.003
		2tertile	3	5.08	45	76.27	11	18.64		
		3tertile	7	10.94	54	84.38	3	4.69		
	*degree*	1tertile	14	19.44	42	58.33	16	22.22	16.3	0.003
		2tertile	3	5.08	45	76.27	11	18.64		
		3tertile	7	10.94	54	84.38	3	4.69		
	*incloseness*	1tertile	14	21.54	35	53.85	16	24.62	22.3	0.001
		2tertile	2	3.13	51	79.69	11	17.19		
		3tertile	8	12.12	55	83.33	3	4.55		
	*outcloseness*	1tertile	14	20.90	37	55.22	16	23.88	19.5	0.001
		2tertile	3	4.69	50	78.13	11	17.19		
		3tertile	7	10.94	54	84.38	3	4.69		
	*betweenness*	1tertile	16	15.69	64	62.75	22	21.57	14.7	0.005
		2tertile	0	0	22	81.48	5	18.52		
		3tertile	14	20.29	39	56.52	16	23.19		
	*eigenvector*	1tertile	2	3.33	47	78.33	11	18.33	22.3	0.001
		2tertile	8	12.12	55	83.33	3	4.55		
		3tertile	14	19.44	42	58.33	16	22.22		

Note: degree (degree of relationships that surround each individual), in/outdegree (received and given relationships), in/out closeness (degree of closeness), betweenness (degree of intermediation), eigenvector (degree of prestige or influence).

## Data Availability

The data presented in this study are available on request from the corresponding author.

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
