# Peer review of "Descriptive Study of Friendship Networks in Adolescents and Their Relationship with Self-Efficacy and Alcohol Consumption Using Social Network Analysis"

_ijerph, 2022, doi:10.3390/ijerph191811527_

Round 1
Reviewer 1 Report
General comments:
Thank you for the opportunity to review this interesting article. For the review of the research I have used the STROBE guidelines. I believe that the topic of the manuscript is interesting and provides new evidence for publication in a journal of this category.
I believe that this manuscript should be improved for publication in IJERPH.
Specific comments:
1. Writing
The writing, structure and organization of the manuscript is in accordance with the guidelines.
2. Title
The title reflects the content and problem studied.
3. Abstract
The abstract reflects reflects the manuscript and provide an informative and balanced summary of what was done and what was found.
line 23 and 24 are not results. Perhaps, it is introduction
The numerical results or at least the effect on the results should be indicated.
The study design should be indicated by a commonly used term in the title or abstract.
4. Key Words
The keywords are representative of the subject studied and exposed. This keywords not are Mesh, alcohol (mesh is Alcohol Drinking), self-efficacy, network, is the same as social network analysis.
5. Background
The background reflects the state of the art in relation to the study. The objective of the study is mentioned, as well as the justification for the choice and importance of studying this theme.
6. Methods
the methodology must be improved.
· They don´t describe the setting, locations, and relevant dates.
· Eligibility criteria and the sources and methods of participant selection is not mentioned.
· The normality test is used for quantitative variables. You only mention using chi-square. Do you need to explain it better?
The variables are correctly explained and you use validated questionnaires.
7. Findings
What was the initial number of participants? You should indicate the reasons for non-participation at each stage, if any, and consider the use of flow charts.
the results are clearly explained and a good report is made of them with tables
8. Discussion
The key results of the discussion are concrete.
In addition, it includes the main strengths and weaknesses in relation to other studies, discussing important differences in the results.
Limitations have been exposed
9. Conclusion.
The conclusion are clear and concise and are in line with the objective.
10. Application to Pratice
The practical application of this investigation is not explained.
11. References
The references used are correct.
You must improve some references like 33 and use the same bibliography format (16,17,18, 28)
Author Response
Thank you very much for the contributions. The improvements made are shown below.:
Summary
The summary has been improved based on the considerations raised. The results have been improved and the abstract has been restructured as can be seen on lines 22 to 29.
Keywords
In line 30, the keywords have been modified, taking into account the MeSH descriptors
Methodology
In line 104, the methodology has been expanded, including the time frame in which our research was carried out, as well as the criteria that led to the selection of centers and participants.
Findings
In lines 165 to 168, the total number of participants has been included, as well as other characteristics.
References
References 16,17,18,28 and 33 have been improved. As well as the rest of the references that have also been reviewed
I would like to thank you for the time dedicated to our research and I hope that the improvements made are of interest to you.
Reviewer 2 Report
I would suggest some additional words regarding limitations e.g.:
-"theoretical model"
- qualitative study
- Alcohol Use Disorders Identification Test (AUDIT) is validated for adults- how is with adolescents...
Author Response
Thank you very much for your contributions, they have served to enrich our research.
In the suggestion "add additional words regarding limitations", on lines 259 to 263 limitations have been added to improve with respect to the suggestions made.
Also consider the use of the Audit as a screening tool in Adolescents. Numerous investigations have verified the adequate psychometric behavior of the AUDIT in the adolescent field. The work of Boubeta, AR et al, in 2017 (Validation of the "Alcohol Consumption Disorder Identification Test" (AUDIT) in a Spanish adolescent population. Psychology Conductual, 25(2), 371.) provides important evidence that the AUDIT is a good tool for screening risky consumption of alcohol in adolescents and may be used from now on in Spain with the required psychometric guarantees and applying contrasted cut-off points.
I would like to thank you for the time dedicated to our research and I hope that the improvements made are of interest to you.
Reviewer 3 Report
The purpose of this study appears to be to investigate whether or not friendship networks are associated with self-efficacy and alcohol consumption. Significant differences in self-efficacy were associated with social network metrics. Secondly, the authors tested the association between self-efficacy and alcohol consumption, concluding that "self-efficacy is a predictor of alcohol consumption."
The underlying assumption appears to be that social networks are predictors of alcohol consumption because they produce self-efficacy. The problem with this line of reasoning is that the study that was conducted is cross-sectional. It cannot be concluded that social networks are "predictors" of alcohol consumption. At most, the study can report an association between network participation and alcohol consumption. It should also be noted that previous research has shown that adolescents who are higher consumers of alcohol and drugs choose the peers with whom to interact. This would suggest individual characteristics such as susceptibility to use of higher quantities of alcohol and drugs, sometimes in association with familial risk for AUD/SUD, may determine the social network rather than the other way around. This literature was not discussed.
There are presentation problems with the manuscript.
(1) The following sentence needs attention for grammar. " We use an online platform with validated self-reported questionnaires was used for data collection."
(2) Table 2 is split between pages making it very difficult to read and interpret. Also, it is unclear if the table at top of page 6 is part of Table 2 or collapsed values based on Table 2 pages 4 and 5.
(3) The statistical analysis presented on page 6 suggests that the probabilities reported may have been influenced by expected values in the Chi Square analysis, resulting in the probability estimates being inaccurate. The table shows 27 different combinations resulting in < 5 cases in many of the cells.
(4) Figure 1 shows the networks graphically and their association with alcohol consumption but in a qualitative rather than quantitative manner. The Figure should be eliminated.
Author Response
Thank you very much for your contributions and your rigor
Regarding the consideration “The underlying assumption appears to be that social networks are predictors of alcohol consumption because they produce self-efficacy” , this has already been improved within the limitations section, as another reviewer suggested. You can check this improvement on the line 259 to 263.
Regarding the following considerations:
- The following sentence needs attention for grammar. " We use an online platform with validated self-reported questionnaires was used for data collection."
We have used the same nomenclature that different colleagues have used to refer to the system used. However, we will check for successive works the existence of different grammatical forms that improve our investigations.
- Table 2 is split between pages making it very difficult to read and interpret. Also, it is unclear if the table at top of page 6 is part of Table 2 or collapsed values based on Table 2 pages 4 and 5.
The table has been placed on two pages, on line 192. Due to its content and breadth, it is totally impossible for us to place it in any other way. We are very sorry for the inconvenience it may cause.
- The statistical analysis presented on page 6 suggests that the probabilities reported may have been influenced by expected values in the Chi Square analysis, resulting in the probability estimates being inaccurate. The table shows 27 different combinations resulting in < 5 cases in many of the cells.
Within our research, the centrality property has been used as a measure for the study of social structures within our group. The classification by means of intensity percentiles and the use of the chi square allowed us to see the meaning of the correlation between our variables. We again appreciate the considerations made and we will take them with great gratitude for future research
- Figure 1 shows the networks graphically and their association with alcohol consumption but in a qualitative rather than quantitative manner. The Figure should be eliminated.
Within the Social Network Analysis, the graphic representation is one of the points of greatest strength. Lozares ( Lozares et al, 1996) argues that the ARS is sometimes presented as a toolbox of a technical-formal and/or graphic nature in the pragmatic and effective search for results without sufficient reflection on the conditions and situations of information collection or on the nature of the data and its contextualization or on the cognitive, factual, dynamic, symbolic dimensions, among others, that the social relationship supposes or on the fields in which said relationships are inserted. This research strategy allows us to identify and weigh the processes investigated and that take place within the classroom, contextualizing the social climate, to provide researchers with the possibility of effectively generating favorable results
I would like to thank you for the time dedicated to our research and I hope that the improvements made are of interest to you.
Round 2
Reviewer 3 Report
(1) Previous Review: The underlying assumption appears to be that social networks are predictors of alcohol consumption because they produce self-efficacy. The problem with this line of reasoning is that the study that was conducted is cross-sectional. It cannot be concluded that social networks are "predictors" of alcohol consumption.
Authors Response: this has already been improved within the limitations section, as another reviewer suggested. You can check this improvement on the line 259 to 263.
Problem at 2nd Review: The sentence in lines 259-253 reads "Self-efficacy is therefore established as an important predictor of facing and withstanding alcohol consumption, contributing to the development of the adolescent's capacities to resist the pressure of the social environment [33]. " The sentence indicates the authors have not accepted the idea that they have not uncovered a predictor of alcohol consumption.
(2) Previous Review Noted: "At most, the study can report an association between network participation and alcohol consumption. It should also be noted that previous research has shown that adolescents who are higher consumers of alcohol and drugs choose the peers with whom to interact. This would suggest individual characteristics such as susceptibility to use of higher quantities of alcohol and drugs, sometimes seem in association with familial risk for AUD/SUD, may determine the social network rather than the other way around. This literature was not discussed."
Problem at 2nd Review: The authors appeared to have ignored this comment completely. There are large scale studies of peer selection and alcohol use such as one based on the Add Health cohort that studied over 2000 teens longitudinally (see Mundt et al BMC Pediatrics, 2012, 12, 115). This paper discusses two different mechanisms in which adolescent alcohol use is influenced by peers. This study noted " However, there is debate over the mechanism by which friends come to resemble one another over time. One possible explanation is that similarities occur as a result of peer influence, or the spread of behaviors and behavioral norms through social ties. In this manner, the behavior of an individual would move toward the average behavior of one’s friends over time. Another pathway to homogeneity within friendships is that friends may be similar due to social selection, or homophily, the tendency for similar people to be attracted to and form friendships among one another." The authors of that study conclude that peer selection plays a major role in alcohol use behavior. The authors of the study under review do not consider this possibility in the interpretation of their data nor do they discuss social selection as possible mechanism. They only consider the first pathway of adolescents moving toward the average behavior of their friends over time.
(3) Previous Review: It was noted that presentation problems with the manuscript. The following sentence needs attention for grammar. " We use an online platform with validated self-reported questionnaires was used for data collection."
Problem at 2nd Review: The authors responded: "We have used the same nomenclature that different colleagues have used to refer to the system used. However, we will check for successive works the existence of different grammatical forms that improve our investigations." It does not appear that they understood that the problem was grammar and not the platform used. The sentence is incorrect because it states "we use" in the same sentence with "was used" so that the tense of the verb needs to corrected to be concordant. Also, rephrasing would be helpful.
(4) Previous Review Noted: Table 2 is split between pages making it very difficult to read and interpret. Also, it is unclear if the table at top of page 6 is part of Table 2 or collapsed values based on Table 2 pages 4 and 5.
Problem at 2nd Review: The authors responded: "The table has been placed on two pages, on line 192. Due to its content and breadth, it is totally impossible for us to place it in any other way. We are very sorry for the inconvenience it may cause." This is not an adequate response as it is not about convenience but clarity of a published manuscript. Table 2 continues to be problematic as the labels on the vertical axis are not readable. If the authors would consider collapsing the groups as suggested under statistical analysis, the table would be smaller and more readable and the statistical analysis would be improved.
(5) Previous Review Noted: The statistical analysis presented on page 6 suggests that the probabilities reported may have been influenced by expected values in the Chi Square analysis, resulting in the probability estimates being inaccurate. The table shows 27 different combinations resulting in < 5 cases in many of the cells.
Problem at 2nd Review: The authors responded: "Within our research, the centrality property has been used as a measure for the study of social structures within our group. The classification by means of intensity percentiles and the use of the chi square allowed us to see the meaning of the correlation between our variables. We again appreciate the considerations made and we will take them with great gratitude for future research." This not an adequate response because the authors do not respond to the statistical problem that results from too few expected cases per cell due to the large number of categories used in their analysis. The results may be spurious. A promise to correct in future research does not justify publication of spurious results. New analyses need to be performed to document the results are statistically correct.
(6) Previous Review Noted: Figure 1 shows the networks graphically and their association with alcohol consumption but in a qualitative rather than quantitative manner. The Figure should be eliminated.
Problem at 2nd Review: The authors responded: "This research strategy allows us to identify and weigh the processes investigated and that take place within the classroom, contextualizing the social climate, to provide researchers with the possibility of effectively generating favorable results." This response does not address the issue of graphical results being presented qualitatively. The figure is not informative about social networks and alcohol consumption based on quantitative estimates.
Author Response
Dear Reviewer,
We appreciate all the considerations provided. After these second comments, we have understood many of our bugs, which we have tried to correct. Thanks to their contributions, our research enjoys greater rigor and meaning..
Improvement suggestion I and II
Sorry for this mistake. The word "predictor" has been replaced by the word "mediator" as you can see in the line. The literature supports the use of the word "mediator" as a facilitator. We have proceeded to review the literature that will help us choose an appropriate term for our results.
- Quiroga Garza, A., Canales Vela, M., Cañamar Decanini, P., De la Peña Zambrano, X., García Puerta, M. J., Moreno Saldaña, S., & Piñeyro Velázquez, A. R. (2022). Predictores de abuso de alcohol en personas adultas con pareja estable. Health & Addictions/Salud y Drogas, 22(2).
- Foster, D. W., Yeung, N., & Neighbors, C. (2014). I think I can't: Drink refusal self-efficacy as a mediator of the relationship between self-reported drinking identity and alcohol use. Addictive behaviors, 39(2), 461-468.
Improvement suggestion III
Sorry again for the mistake. We do not visualize that failure of verb tenses. It has been modified on the line 19
Improvement suggestion IV
We have proceeded to elaborate the tables with visual improvements. In this way the data is more organized and readable for the reader. You can check it on line 194. We greatly appreciate this aspect, since it has meant a substantial improvement for the reading and understanding of our work.
Improvement suggestion V
We consider that the probability estimates due to the low number of cases may be inaccurate, for this reason we have included this aspect within our limitations. You can check these modifications on line 333.
We would like to be able to explain that within the SNA it is possible to sample a complete network (in which the dependency of the data does not allow probabilistic sampling) but also the use of centrality measures that are more reliable in low or high data contexts. incomplete.
This old but important article shows that indegree especially is very reliable, even in extreme contexts of missing data:
- Costenbader, E., & Valente, T.W. (2003). The stability of centrality measures when networks are sampled. Social Networks, 25(4), 283–307. https://doi.org/10.1016/S0378-8733(03)00012-1
Improvement suggestion VI
Indeed, the presentation of a graph is not scientifically valuable. We have used your presentation to point out a certain effect also a result of the statistical analysis. The use of the graph allows non-SNA expert readers to clarify results. For this reason, we ask that if you like it, consider maintaining the graph to give clarity to our investigation.